# Stochastic Ratio Matching of RBMs for Sparse High-Dimensional Inputs

**Yann N. Dauphin,  Yoshua Bengio**
Département d'informatique et de recherche opérationnelle
Université de Montréal
Montréal, QC H3C 3J7
`dauphiya@iro.umontreal.ca`,
`Yoshua.Bengio@umontreal.ca`

## Abstract

Sparse high-dimensional data vectors are common in many application domains where a very large number of rarely non-zero features can be devised. Unfortunately, this creates a computational bottleneck for unsupervised feature learning algorithms such as those based on auto-encoders and RBMs, because they involve a reconstruction step where the whole input vector is predicted from the current feature values. An algorithm was recently developed to successfully handle the case of auto-encoders, based on an importance sampling scheme stochastically selecting which input elements to actually reconstruct during training for each particular example. To generalize this idea to RBMs, we propose a stochastic ratio-matching algorithm that inherits all the computational advantages and un-biasedness of the importance sampling scheme. We show that stochastic ratio matching is a good estimator, allowing the approach to beat the state-of-the-art on two bag-of-word text classification benchmarks (20 Newsgroups and RCV1), while keeping computational cost linear in the number of non-zeros.

## 1   Introduction

Unsupervised feature learning algorithms have recently attracted much attention, with the promise of letting the data guide the discovery of good representations. In particular, unsupervised feature learning is an important component of many Deep Learning algorithms (Bengio, 2009), such as those based on auto-encoders (Bengio *et al.*, 2007) and Restricted Boltzmann Machines or RBMs (Hinton *et al.*, 2006). Deep Learning of representations involves the discovery of several levels of representation, with some algorithms able to exploit unlabeled examples and unsupervised or semi-supervised learning.

Whereas Deep Learning has mostly been applied to computer vision and speech recognition, an important set of application areas involve *high-dimensional sparse input vectors*, for example in some Natural Language Processing tasks (such as the text categorization tasks tackled here), as well as in information retrieval and other web-related applications where a very large number of rarely non-zero features can be devised. We would like learning algorithms whose computational requirements grow with the number of non-zeros in the input but not with the total number of features. Unfortunately, auto-encoders and RBMs are computationally inconvenient when it comes to handling such high-dimensional sparse input vectors, because they require a form of *reconstruction* of the input vector, *for all the elements of the input vector, even the ones that were zero*.

In Section 2, we recapitulate the Reconstruction Sampling algorithm (Dauphin *et al.*, 2011) that was proposed to handle that problem in the case of auto-encoder variants. The basic idea is to use an

importance sampling scheme to stochastically select a subset of the input elements to reconstruct, and importance weights to obtain an unbiased estimator of the reconstruction error gradient.

In this paper, we are interested in extending these ideas to the realm of RBMs. In Section 3 we briefly review the basics of RBMs and the Gibbs chain involved in training them. Ratio matching (Hyvärinen, 2007), is an inductive principle and training criterion that can be applied to train RBMs but does not require a Gibbs chain. In Section 4, we present and justify a novel algorithm based on ratio matching order to achieve our objective of taking advantage of highly sparse inputs. The new algorithm is called *Stochastic Ratio Matching* or SRM. In Section 6 we present a wide array of experimental results demonstrating the successful application of Stochastic Ratio Matching, both in terms of computational performance (flat growth of computation as the number of non-zeros is increased, linear speedup with respect to regular training) and in terms of generalization performance: the state-of-the-art on two text classification benchmarks is achieved and surpassed. An interesting and unexpected result is that we find the *biased version* of the algorithm (without reweighting) to yield more discriminant features.

## 2 Reconstruction Sampling

An auto-encoder learns an encoder function $f$ mapping inputs $x$ to features $h = f(x)$, and a decoding or reconstruction function $g$ such that $g(f(x)) \approx x$ for training examples $x$. See Bengio *et al.* (2012) for a review. In particular, with the denoising auto-encoder, $x$ is stochastically corrupted into $\tilde{x}$ (e.g. by flipping some bits) and trained to make $g(f(\tilde{x})) \approx x$. To avoid the expensive reconstruction $g(h)$ when the input is very high-dimensional, Dauphin *et al.* (2011) propose that for each example, a small random subset of the input elements be selected for which $g_i(h)$ and the associated reconstruction error is computed. To make the corresponding estimator of reconstruction error (and its gradient) unbiased, they propose to use an importance weighting scheme whereby the loss on the $i$-th input is weighted by the inverse of the probability that it be selected. To reduce the variance of the estimator, they propose to always reconstruct the $i$-th input if it was one of the non-zeros in $x$ or in $\tilde{x}$, and to choose uniformly at random an equal number of zero elements. They show that the unbiased estimator yields the expected linear speedup in training time compared to the deterministic gradient computation, while maintaining good performance for unsupervised feature learning. We would like to extend similar ideas to RBMs.

## 3 Restricted Boltzmann Machines

A restricted Boltzmann machine (RBM) is an undirected graphical model with binary variables (Hinton *et al.*, 2006): observed variables $\mathbf{x}$ and hidden variables $\mathbf{h}$. In this model, the hidden variables help uncover higher order correlations in the data.

The energy takes the form
$$-E(\mathbf{x}, \mathbf{h}) = \mathbf{h}^T \mathbf{W} \mathbf{x} + \mathbf{b}^T \mathbf{h} + \mathbf{c}^T \mathbf{x}$$
with parameters $\theta = (\mathbf{W}, \mathbf{b}, \mathbf{c})$.

The RBM can be trained by following the gradient of the negative log-likelihood
$$-\frac{\partial \log P(\mathbf{x})}{\partial \theta} = E_{data}\left[\frac{\partial F(\mathbf{x})}{\partial \theta}\right] - E_{model}\left[\frac{\partial F(\mathbf{x})}{\partial \theta}\right]$$

where $F(x)$ is the free energy (unnormalized log-probability associated with $P(x)$). However, this gradient is intractable because the second expectation is combinatorial. Stochastic Maximum Likelihood or SML (Younes, 1999; Tieleman, 2008) estimates this expectation using sample averages taken from a persistent MCMC chain (Tieleman, 2008). Starting from $\mathbf{x}^i$ a step in this chain is taken by sampling $\mathbf{h}^i \sim P(\mathbf{h}|\mathbf{x}^i)$, then we have $\mathbf{x}^{i+1} \sim P(\mathbf{x}|\mathbf{h}^i)$. SML-$k$ is the variant where $k$ is the number of steps between parameter updates, with SML-1 being the simplest and most common choice, although better results (at greater computational expense) can be achieved with more steps.

Training the RBM using SML-1 is on the order of $O(dn)$ where $d$ is the dimension of the input variables and $n$ is the number of hidden variables. In the case of *high-dimensional sparse vectors* with $p$ non-zeros, SML does not take advantage of the sparsity. More precisely, sampling $P(\mathbf{h}|\mathbf{x})$

(inference) can take advantage of sparsity and costs $O(pn)$ computations while "reconstruction", i.e., sampling from $P(\mathbf{x}|\mathbf{h})$ requires $O(dn)$ computations. Thus scaling to larger input sizes $n$ yields a linear increase in training time even if the number of non-zeros $p$ in the input remains constant.

## 4  Ratio Matching

Ratio matching (Hyvärinen, 2007) is an estimation method for statistical models where the normalization constant is not known. It is similar to score matching (Hyvärinen, 2005) but applied on discrete data whereas score matching is limited to continuous inputs, and both are computationally simple and yield consistent estimators. The use of Ratio Matching in RBMs is of particular interest because their normalization constant is computationally intractable.

The core idea of ratio matching is to match ratios of probabilities between the data and the model. Thus Hyvärinen (2007) proposes to minimize the following objective function

$$
P_x(\mathbf{x}) \sum_{i=1}^{d} \left[ g\left( \frac{P_x(\mathbf{x})}{P_x(\bar{\mathbf{x}}^i)} \right) - g\left( \frac{P(\mathbf{x})}{P(\bar{\mathbf{x}}^i)} \right) \right]^2 + \left[ g\left( \frac{P_x(\bar{\mathbf{x}}^i)}{P_x(\mathbf{x})} \right) - g\left( \frac{P(\bar{\mathbf{x}}^i)}{P(\mathbf{x})} \right) \right]^2 \tag{1}
$$

where $P_x$ is the true probability distribution, $P$ the distribution defined by the model, $g(x) = \frac{1}{1+x}$ is an activation function and $\bar{\mathbf{x}}^i = (x_1, x_2, \ldots, 1 - x_i, \ldots, x_d)$. In this form, we can see the similarity between score matching and ratio matching. The normalization constant is canceled because $\frac{P(\mathbf{x})}{P(\bar{\mathbf{x}}^i)} = \frac{e^{-F(\mathbf{x})}}{e^{-F(\bar{\mathbf{x}}^i)}}$, however this objective requires access to the true distribution $P_x$ which is rarely available.

Hyvärinen (2007) shows that the Ratio Matching (RM) objective can be simplified to

$$
J_{RM}(\mathbf{x}) = \sum_{i=1}^{d} \left( g\left( \frac{P(\mathbf{x})}{P(\bar{\mathbf{x}}^i)} \right) \right)^2 \tag{2}
$$

which does not require knowledge of the true distribution $P_x$. This objective can be described as ensuring that the training example $\mathbf{x}$ has the highest probability in the neighborhood of points at hamming distance 1.

We propose to rewrite Eq. 2 in a form reminiscent of auto-encoders:

$$
J_{RM}(\mathbf{x}) = \sum_{i=1}^{d} (x_i - P(x_i = 1|\mathbf{x}_{-i}))^2. \tag{3}
$$

This will be useful for reasoning about this estimator. The main difference with auto-encoders is that each input variable is predicted by excluding it from the input.

Applying Equation 2 to the RBM we obtain $J_{RM}(\mathbf{x}) = \sum_{i=1}^{d} \left( \sigma(F(\mathbf{x}) - F(\bar{\mathbf{x}}^i)) \right)^2$. The gradients have the familiar form

$$
-\frac{\partial J_{RM}(\mathbf{x})}{\partial \theta} = \sum_{i=1}^{d} 2\eta_i \left[ \frac{\partial F(\mathbf{x})}{\partial \theta} - \frac{\partial F(\bar{\mathbf{x}}^i)}{\partial \theta} \right] \tag{4}
$$

with $\eta_i = \left( \sigma(F(\mathbf{x}) - F(\bar{\mathbf{x}}^i)) \right)^2 - \left( \sigma(F(\mathbf{x}) - F(\bar{\mathbf{x}}^i)) \right)^3$.

A naive implementation of this objective is $O(d^2n)$ because it requires $d$ computations of the free energy per example. This is much more expensive than SML as noted by Marlin *et al.* (2010). Thankfully, as we argue here, it is possible to greatly reduce this complexity by reusing computation and taking advantage of the parametrization of RBMs. This can be done by saving the results of the computations $\alpha = \mathbf{c}^T \mathbf{x}$ and $\beta_j = \sum_i W_{ji} x_i + b_j$ when computing $F(\mathbf{x})$. The computation of $F(\bar{\mathbf{x}}^i)$ can be reduced to $O(n)$ with the formula $-F(\bar{\mathbf{x}}^i) = \alpha - (2x_i - 1)c_i + \sum_j \log(1 + e^{\beta_j - (2x_i - 1)W_{ji}})$. This implementation is $O(dn)$ which is the same complexity as SML. However, like SML, RM does not take advantage of sparsity in the input.

# 5 Stochastic Ratio Matching

We propose Stochastic Ratio Matching (SRM) as a more efficient form of ratio matching for high-dimensional sparse distributions. The ratio matching objective requires the summation of $d$ terms in $O(n)$. The basic idea of SRM is to estimate this sum using a very small fraction of the terms, randomly chosen. If we rewrite the ratio matching objective as an expectation over a discrete distribution

$$J_{RM}(\mathbf{x}) = d \sum_{i=1}^{d} \frac{1}{d} g^2 \left( \frac{P(\mathbf{x})}{P(\bar{\mathbf{x}}^i)} \right) = dE \left[ g^2 \left( \frac{P(\mathbf{x})}{P(\bar{\mathbf{x}}^i)} \right) \right] \qquad (5)$$

we can use Monte Carlo methods to estimate $J_{RM}$ without computing all the terms in Equation 2. However, in practice this estimator has a high variance. Thus it is a poor estimator, especially if we want to use very few Monte Carlo samples. The solution proposed for SRM is to use an Importance Sampling scheme to obtain a lower variance estimator of $J_{RM}$. Combining Monte Carlo with importance sampling, we obtain the SRM objective

$$J_{SRM}(\mathbf{x}) = \sum_{i=1}^{d} \frac{\gamma_i}{E[\gamma_i]} g^2 \left( \frac{P(\mathbf{x})}{P(\bar{\mathbf{x}}^i)} \right) \qquad (6)$$

where $\gamma_i \sim P(\gamma_i = 1|\mathbf{x})$ is the so-called proposal distribution of our importance sampling scheme. The proposal distribution determines which terms will be used to estimate the objective since only the terms where $\gamma_i = 1$ are non-zero. $J_{SRM}(\mathbf{x})$ is an unbiased estimator of $J_{RM}(\mathbf{x})$, i.e.,

$$\begin{aligned} E[J_{SRM}(\mathbf{x})] &= \sum_{i=1}^{d} \frac{E[\gamma_i]}{E[\gamma_i]} g^2 \left( \frac{P(\mathbf{x})}{P(\bar{\mathbf{x}}^i)} \right) \\ &= J_{RM}(\mathbf{x}) \end{aligned}$$

The intuition behind importance sampling is that the variance of the estimator can be reduced by focusing sampling on the largest terms of the expectation. More precisely, it is possible to show that the variance of the estimator is minimized when $P(\gamma_i = 1|\mathbf{x}) \propto g^2(P(\mathbf{x})/P(\bar{\mathbf{x}}^i))$. Thus we would like the probability $P(\gamma_i = 1|\mathbf{x})$ to reflect how large the error $(x_i - P(x_i = 1|\mathbf{x}_{-i}))^2$ will be. The challenge is finding a good approximation for $(x_i - P(x_i = 1|\mathbf{x}_{-i}))^2$ and to define a proposal distribution that is efficient to sample from.

Following Dauphin *et al.* (2011), we propose such a distribution for high-dimensional sparse distributions. In these types of distributions the marginals $P_x(x_i = 1)$ are very small. They can easily be learned by the biases $\mathbf{c}$ of the model, and may even be initialized very close to their optimal value. Once the marginals are learned, the model will likely only make wrong predictions when $P_x(x_i = 1|\mathbf{x}_{-i})$ differs significantly from $P_x(x_i = 1)$. If $x_i = 0$ then the error $(0 - P(x_i = 1|\mathbf{x}_{-i}))^2$ is likely small because the model has a high bias towards $P(x_i = 0)$. Conversely, the error will be high when $x_i = 1$. In other words, the model will mostly make errors for terms where $x_i = 1$ and a small number of dimensions where $x_i = 0$. We can use this to define the heuristic proposal distribution

$$P(\gamma_i = 1|\mathbf{x}) = \begin{cases} 1 & \text{if } x_i = 1 \\ p/(d - \sum_j 1_{x_j > 0}) & otherwise \end{cases} \qquad (7)$$

where $p$ is the average number of non-zeros in the data. The idea is to always sample the terms where $x_i = 1$ and a subset of $k$ of the $(d - \sum_j 1_{x_j > 0})$ remaining terms where $x_i = 0$. Note that if we sampled the $\gamma_i$ independently, we would get $E[k] = p$.

However, instead of sampling those $\gamma_i$ bits independently, we find that much smaller variance is obtained by sampling a number of zeros $k$ that is **constant** for all examples, i.e., $k = p$. A random $k$ can cause very significant variance in the gradients and this makes stochastic gradient descent more difficult. In our experiments we set $k = p = E[\sum_j 1_{x_j > 0}]$ which is a small number by definition of these sparse distributions, and guarantees that computation costs will remain constant as $n$ increases for a fixed number of non-zeros. The computational cost of SRM per training example is $O(pn)$, as opposed to $O(dn)$ for RM. While simple, we find that this heuristic proposal distribution works well in practice, as shown below.

For comparison, we also perform experiments with a biased version of Equation 6

$$J_{BiasedSRM}(\mathbf{x}) = \sum_{i=1}^{d} \gamma_i g^2 \left( \frac{P(\mathbf{x})}{P(\bar{\mathbf{x}}^i)} \right).$$  (8)

This will allow us to gauge the effectiveness of our importance weights for unbiasing the objective. The biased objective can be thought as down-weighting the ratios where $x_i = 0$ by a factor of $E[\gamma_i]$.

SRM is related to previous work (Dahl *et al.*, 2012) on applying RBMs to high-dimensional sparse inputs, more precisely multinomial observations, e.g., one K-ary multinomial for each word in an n-gram window. A careful choice of Metropolis-Hastings transitions replaces Gibbs transitions and allows to handle large vocabularies. In comparison, SRM is geared towards general sparse vectors and involves an extremely simple procedure without MCMC.

## 6  Experimental Results

In this section, we demonstrate the effectiveness of SRM for training RBMs. Additionally, we show that RBMs are useful features extractors for topic classification.

**Datasets**   We have performed experiments with the Reuters Corpus Volume I (RCV1) and 20 Newsgroups (20 NG). RCV1 is a benchmark for document classification of over 800,000 news wire stories (Lewis *et al.*, 2004). The documents are represented as bag-of-words vectors with 47,236 dimensions. The training set contains 23,149 documents and the test set has 781,265. While there are 3 types of labels for the documents, we focus on the task of predicting the topic. There are a set of 103 non-mutually exclusive topics for a document. We report the performance using the $F_{1.0}$ measure for comparison with the state of the art. 20 Newsgroups is a collection of Usenet posts composing a training set of 11,269 examples and 7505 test examples. The bag-of-words vectors contain 61,188 dimensions. The postings are to be classified into one of 20 categories. We use the *by-date* train/test split which ensures that the training set contains postings preceding the test examples in time. Following Larochelle *et al.* (2012), we report the classification error and for a fair comparison we use the same preprocessing[1].

**Methodology**   We compare the different estimation methods for the RBM based on the log-likelihoods they achieve. To do this we use Annealed Importance Sampling or AIS (Salakhutdinov and Murray, 2008). For all models we average 100 AIS runs with 10,000 uniformly spaced reverse temperatures $\beta_k$. We compare RBMs trained with ratio matching, stochastic ratio matching and biased stochastic ratio matching. We include experiments with RBMs trained with SML-1 for comparison.

Additionally, we provide experiments to motivate the use of high-dimensional RBMs in NLP. We use the RBM to pretrain the hidden layers of a feed-forward neural network (Hinton *et al.*, 2006). This acts as a regularization for the network and it helps optimization by initializing the network close to a good local minimum (Erhan *et al.*, 2010).

The hyper-parameters are cross-validated on a validation set consisting of 5% of the training set. In our experiments with AIS, we use the validation log-likelihood as the objective. For classification, we use the discriminative performance on the validation set. The hyper-parameters are found using random search (Bergstra and Bengio, 2012) with 64 trials per set of experiments. The learning rate for the RBMs is sampled from $10^{-[0,3]}$, the number of hidden units from $[500, 2000]$ and the number of training epochs from $[5, 20]$. The learning rate for the MLP is sampled from $10^{-[2,0]}$. It is trained for 32 epochs using early-stopping based on the validation set. We regularize the MLP by dropping out 50% of the hidden units during training (Hinton *et al.*, 2012). We adapt the learning rate dynamically by multiplying it by 0.95 when the validation error increases.

All experiments are run on a cluster of double quad-core Intel Xeon E5345 running at 2.33Ghz with 2GB of RAM.

Table 1: Log-probabilities estimated by AIS for the RBMs trained with the different estimation methods. With a fixed budget of epochs, SRM achieves likelihoods on the test set comparable with RM and SML-1.

| | | ESTIMATES | | AVG. LOG-PROB. | |
|---|---|---|---|---|---|
| | | $\log \hat{Z}$ | $\log(\hat{Z} \pm \hat{\sigma})$ | TRAIN | TEST |
| RCV1 | BIASED SRM | 1084.96 | 1079.66, 1085.65 | -758.73 | -793.20 |
| | SRM | 325.26 | 325.24, 325.27 | -139.79 | -151.30 |
| | RM | 499.88 | 499.48, 500.17 | -119.98 | **-147.32** |
| | SML-1 | 323.33 | 320.69, 323.99 | -138.90 | -153.50 |
| 20 NG | BIASED SRM | 1723.94 | 1718.65, 1724.63 | -960.34 | -1018.73 |
| | SRM | 546.52 | 546.55, 546.49 | -178.39 | -190.72 |
| | RM | 975.42 | 975.62, 975.18 | -159.92 | **-185.61** |
| | SML-1 | 612.15 | 611.68, 612.46 | -173.56 | -188.82 |

## 6.1 Using SRM to train RBMs

We can measure the effectiveness of SRM by comparing it with various estimation methods for the RBM. As the RBM is a generative model, we must compare these methods based on the log-likelihoods they achieve. Note that Dauphin *et al.* (2011) relies on the classification error because there is no accepted performance measure for DAEs. As both RM and SML scale badly with input dimension, we restrict the dimension of the dataset to the $p = 1,000$ most frequent words. We will describe experiments with all dimensions in the next section.

As seen in Table 1, SRM is a good estimator for training RBMs and is a good approximation of RM. We see that with the same budget of epochs SRM achieves log-likelihoods comparable with RM on both datasets. The striking difference of more than 500 nats with Biased SRM shows that the importance weights successfully unbias the estimator. Interestingly, we observe that RM is able to learn better generative models than SML-1 for both datasets. This is similar to Marlin *et al.* (2010) where Pseudolikelihood achieves better log-likelihood than SML on a subset of 20 newsgroups. We observe this is an optimization problem since the training log-likelihood is also higher than RM. One explanation is that SML-1 might experience mixing problems (Bengio *et al.*, 2013).

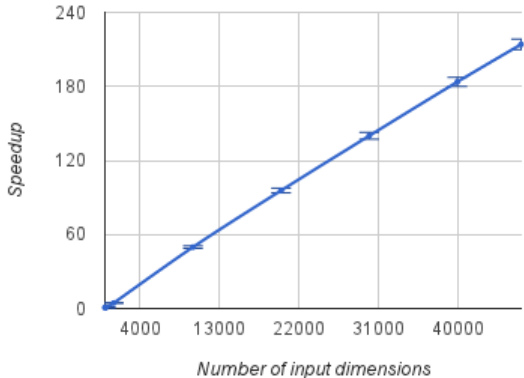

Figure 1: Average speedup in the calculation of gradients by using the *SRM* objective compared to *RM*. The speed-up is linear and reaches up to 2 orders of magnitude.

Figure 1 shows that as expected SRM achieves a linear speed-up compared to RM, reaching speed-ups of 2 orders of magnitude. In fact, we observed that the computation time of the gradients for RM scale linearly with the size of the input while the computation time of SRM remains fairly constant because the number of non-zeros varies little. This is an important property of SRM which makes it suitable for very large scale inputs.

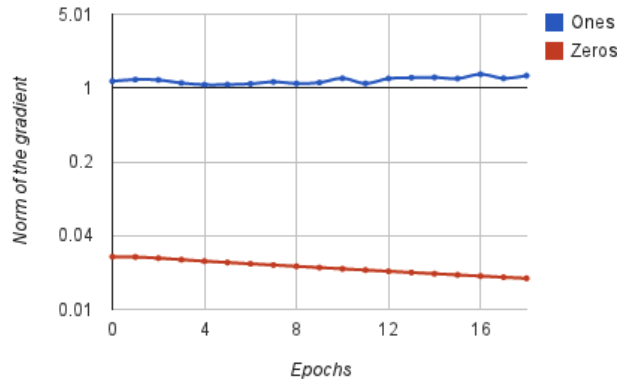

Figure 2: Average norm of the gradients for the terms in Equation 2 where $x_i = 1$ and $x_i = 0$. Confirming the hypothesis for the proposal distribution the terms where $x_i = 1$ are 2 orders of magnitude larger.

The importance sampling scheme of SRM (Equation 7) relies on the hypothesis that terms where $x_i = 1$ produce a larger gradient than terms where $x_i = 0$. We can verify this by monitoring the average gradients during learning on RCV1. Figure 2 demonstrates that the average gradients for the terms where $x_i = 1$ is 2 orders of magnitudes larger than those where $x_i = 0$. This confirms the hypothesis underlying the sampling scheme of SRM.

## 6.2 Using RBMs as feature extractors for NLP

Having established that SRM is an efficient unbiased estimator of RM, we turn to the task of using RBMs not as generative models but as feature extractors. We find that keeping the bias in SRM is helpful for classification. This is similar to the known result that contrastive divergence, which is biased, yields better classification results than persistent contrastive divergence, which is unbiased. The bias increases the weight of non-zeros features. The superior performance of the biased objective suggests that the non-zero features contain more information about the classification task. In other words, for these tasks it's more important to focus on what is there than what is not there.

Table 2: Classification results on RCV1 with all 47,326 dimensions. The DBN trained with SRM achieves state-of-the-art performance.

| MODEL | TEST SET F1 |
|---|---|
| ROCCHIO | 0.693 |
| $k$-NN | 0.765 |
| SVM | 0.816 |
| SDA-MLP (REC. SAMPLING) | 0.831 |
| RBM-MLP (UNBIASED SRM) | 0.816 |
| RBM-MLP (BIASED SRM) | 0.829 |
| DBN-MLP (BIASED SRM) | **0.836** |

On RCV1, we train our models on all 47,326 dimensions. The RBM trained with SRM improves on the state-of-the-art (Lewis *et al.*, 2004), as shown in Table 2. The total training time for this RBM using SRM is 57 minutes. We also train a Deep Belief Net (DBN) by stacking an RBM trained with SML on top of the RBMs learned with SRM. This type of 2-layer deep architecture is able to significantly improve the performance on that task (Table 2). In particular the DBN does significantly better than a stack of denoising auto-encoders we trained using biased reconstruction sampling (Dauphin *et al.*, 2011), which appears as *SDA-MLP (Rec. Sampling)* in Table 2.

We apply RBMs trained with SRM on 20 newsgroups with all 61,188 dimensions. We see in Table 3 that this approach improves the previous state-of-the-art by over 1% (Larochelle *et al.*, 2012), beating non-pretrained MLPs and SVMs by close to 10 %. This result is closely followed by the DAE trained with reconstruction sampling which in our experiments reaches 20.6% test error. The

Table 3: Classification results on 20 Newsgroups with all 61,188 dimensions. Prior results from (Larochelle *et al.*, 2012). The RBM trained with SRM achieves state-of-the-art results.

| MODEL | TEST SET ERROR |
|---|---|
| SVM | 32.8 % |
| MLP | 28.2 % |
| RBM | 24.9 % |
| HDRBM | 21.9 % |
| DAE-MLP (REC. SAMPLING) | 20.6 % |
| RBM-MLP (BIASED SRM) | **20.5** % |

simpler RBM trained by SRM is able to beat the more powerful HD-RBM model because it uses all the 61,188 dimensions.

## 7 Conclusion

We have proposed a very simple algorithm called Stochastic Ratio Matching (SRM) to take advantage of sparsity in high-dimensional data when training discrete RBMs. It can be used to estimate gradients in $O(np)$ computation where $p$ is the number of non-zeros, yielding linear speedup against the $O(nd)$ of Ratio Matching (RM) where $d$ is the input size. It does so while providing an unbiased estimator of the ratio matching gradient. Using this efficient estimator we train RBMs as features extractors and achieve state-of-the-art results on 2 text classification benchmarks.

## Footnotes

[1]http://qwone.com/˜jason/20Newsgroups/20news-bydate-matlab.tgz

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
