[Reviews · NeurIPS 2013]

Submitted by Assigned_Reviewer_4

The paper uses a subsampling-based method to speed up ratio matching training of
RBMs on high-dimensional sparse binary data. The proposed approach is a simple
adaptation of the method proposed by Dauphin et al. (2011) for denoising
autoencoders.

The paper is well written, with both the background and the method clearly
described. The first part of the experimental section, dealing with training
RBMs as generative models is convincing, though Table 1 would benefit from the
addition of the training times for the methods. The feature learning
experiments are somewhat less informative. It would be helpful to report the
performance of models trained using (unbiased) SRM, even if it is not
competitive. It is also unclear how the reconstruction sampling results were
obtained? Did the authors implement that method themselves?

Were the hyperparameters really chosen using cross-validation as stated on l.
250? After all, using a 5% validation set would have required training the
models 20 times.

Based on the reported results, only the biased version of the proposed method
seems to be on par with (or very marginally superior to) reconstruction
sampling for feature learning. The authors however do not provide a convincing
explanation for the better performance of the biased algorithm, which is
unfortunate because this finding is perhaps the most interesting contribution
of the paper. Could it be simply that not dividing by E[gamma_i] allows much
larger learning rates to be used?

Finally, the paper fails to mention that the RM speedup trick of caching
intermediate computation results was already noted and implemented by Marlin et
al. (2010).

Some suggestions:
l. 061: flat growth -> linear growth
l. 103: is on the order -> takes time on the order
Eq. 1: Brackets around the two terms inside the sum
Eq. 4: There should be no "-" on the LHS, as this is the gradient, not
the parameter update.
Summary: The paper adapts a subsampling trick introduced for training autoencoders on
sparse high-dimensional data to RBMs. The paper is fairly well executed but
incremental.

Submitted by Assigned_Reviewer_5

This paper solves the computational problem of training RBMs on high-dimensional sparse data by combining the ratio matching algorithm with importance sampling algorithms used to solve the analogous problem for denoising auto-encoders. Experimental results using RBMs on bag-of-words text classification tasks demonstrate the success of the training algorithm.

The paper is clear and solves an important problem that paves the way for new Boltzmann machine based models of text data. Although most of the pieces of this solution were already present in the literature, this paper describes a novel combination of ratio matching for RBM training and importance sampling with a non-zero input dimension heuristic. There are no fundamental problems with the paper as is, but there are a few sections that could be improved. The results for unbiased SRM should really be included in the tables. It would be worrying if unbiased SRM learned *much* worse feature detectors although obviously being slightly worse is expected. The text of section 6.2 should be revised to clarify for those not intimately familiar with Larochelle et al. 2012 that the authors of the submission ran the reconstruction sampling experiments.

Another task that would be interesting to use RBMs trained with stochastic ratio matching on would be modeling the joint distribution of words in an n-gram. With a large vocabulary, there would be an extreme level of sparsity from the large categorical distributions beyond the sparsity of any of the datasets used in the submission. The importance sampling proposal distribution heuristic might break down since the distribution of reconstruction error across dimensions with zero input should have more interesting structure than for bag of words tasks.
Summary: The paper is clear and solves an important problem that paves the way for new Boltzmann machine based models of text data.

Submitted by Assigned_Reviewer_6

Summary:

This paper develops an algorithm that can successfully train RBMs on very high dimensional but sparse input data, such as often arises in NLP problems. The algorithm adapts a previous method developed for denoising autoencoders for use with RBMs. The authors present extensive experimental results verifying that their method learns a good generative model; provides unbiased gradient estimates; attains a two order of magnitude speed up on large sparse problems relative to the standard implementation; and yields state of the art performance on a number of NLP tasks. They also document the curious result that using a biased version of their estimator in fact leads to better performance on the classification tasks they tested.

Pros:

While the underlying idea is present in previous work with autoencoders, transferring this idea to the RBM setting required some non-trivial insights, particularly into the ratio-matching objective for training RBMs.

Every claim is supported by an appropriate figure (learns good generative models; gradient estimates are unbiased; importance sampler proposal distribution selects large-gradient samples; attains a large speed up relative to standard implementation; yields state of the art performance on certain NLP tasks). The experiments have been done to a high standard.

The paper includes some useful other results, including a highly intuitive form for the ratio matching objective, and a method for saving intermediate variables that allows more computationally efficient computation of the free energy terms required by the ratio matching objective.

The paper is very clearly written.

Cons:

While the proposed method does achieve state-of-the-art results, its performance is highly similar to that of the denoising autoencoder. There is no real case made in the paper that the RBM algorithm is particularly more useful than the already developed DAE version of the algorithm.

The experimental results do not give quantitative classification performance results for the unbiased SRM algorithm (e.g. Tables 2-3). While it is stated in the text that the unbiased method was not the top performer, it would still be informative to get a sense of the magnitude of the difference. It would also be valuable to clarify if the denoising autoencoders were trained with or without biased gradient estimates. Since biased gradients work better for the RBM, it seems likely this is true of the autoencoder as well.
Summary: A well-executed but incremental extension of an autoencoder training method for sparse input data to the RBM setting.
Author Feedback

Author rebuttal: Thanks for all the noted typos and required clarifications, which will be dealt with in the revision.

Rev4

"Table 1 would benefit from the addition of the training times for the methods": We will include training times.

"using cross-validation": a single training/validation split was used.

"convincing explanation for the better performance of the biased algorithm":
This came as a surprise to us and would probably be surprising for many. However, a reasonable explanatory hypothesis (which we will add in the paper) is that for this dataset (and maybe for others), the features that are more often non-zero also contain more information about target classification task. Learning rate is not an explanation since it was optimized separately (on the validation set) for each experimental setting.

"It is also unclear how the reconstruction sampling results were obtained? Did the authors implement that method themselves?":
We will clarify in the paper that we ran those experiments.

Marlin et al 2010: the mention of the speedup trick will be noted.

Rev5

Results for unbiased SRM will be added to the tables.

6.2 will be clarified as suggested.

The idea of trying to learn n-gram joint distributions seems good, thank you.

Rev6

"While the proposed method does achieve state-of-the-art results, its performance is highly similar to that of the denoising autoencoder. There is no real case made in the paper that the RBM algorithm is particularly more useful than the already developed DAE version of the algorithm. The paper is therefore largely incremental." :
We disagree that the fact that results are similar to those obtained with DAEs implies that the paper is incremental. There are several reasons why one could prefer to use RBMs rather than DAEs, so allowing RBMs to efficiently be trained on high-dimensional sparse input solves an important problem for this class of learning algorithms.

Results with the unbiased SRM algorithm will be added to the table.

Denoising auto-encoders were trained with a biased estimate of the gradient. This information will be added, as well as results for the unbiased gradient.